

# What 'unexplored' means: mapping regions with digitized natural history records to look for 'biodiversity blindspots'

Laymon Ball[1], Sheila Rodríguez-Machado[1,2], Diego Paredes-Burneo[1,3], Samantha Rutledge[1,2], David A. Boyd[2], David Vander Pluym[1,2], Spenser Babb-Biernacki[1,2], Austin S. Chipps[1,2], Rafet Ç. Öztürk[2,4], Yahya Terzi[2,4] and Prosanta Chakrabarty[1,2,5,6,7]

[1] Department of Biological Sciences, Louisiana State University, Baton Rouge, Louisiana, United States
[2] Museum of Natural Science, Department of Biological Sciences, Louisiana State University, Baton Rouge, LA, United States
[3] Departamento de Dicotiledóneas, Museo de Historia Natural UNMSM, Lima, Peru
[4] Department of Fisheries Technology Engineering, Faculty of Marine Sciences, Karadeniz Technical University, Trabzon, Türkiye
[5] American Museum of Natural History, New York, NY, United States
[6] Smithsonian National Museum of Natural History, Washington, DC, United States
[7] Canadian Museum of Nature, Ottawa, ON, Canada

Corresponding author
Prosanta Chakrabarty,
prosanta@lsu.edu

## ABSTRACT

We examined global records of accessible natural history voucher collections (with publicly available data and reliable locality data) for terrestrial and freshwater vascular plants, fungi, freshwater fishes, birds, mammals, and herpetofauna (amphibians and reptiles) and highlight areas of the world that would be considered undersampled and sometimes called 'unexplored' (*i.e.*, have relatively low, or no evidence of, past sampling efforts) under typical Western-scientific descriptions. We also question what 'unexplored' may mean in these contexts and explain how replacing the term in favor of more nuanced phrasing (*e.g.*, 'biodiversity blindspots,' which emphasizes the lack of publicly available data about specimens) can mitigate future misunderstandings of natural history science. We also highlight geographic regions where there are relatively few or no publicly available natural history records to raise awareness about habitats that might be worthy of future natural history research and conservation. A major finding is that many of the areas that appear 'unexplored' may be in countries whose collections are not digitized (*i.e.*, they don't have metadata such as GPS coordinates about their voucher specimens publicly available). We call for museums to prioritize digitizing those collections from these 'biodiversity blindspots' and for increased funding for museums to aid in these efforts. We also argue for increased scientific infrastructure so that more reference collections with vouchers can be kept in the countries of origin (particularly those countries lacking such infrastructure currently).

## INTRODUCTION

Collections-based natural history research is the work of observing, securing, obtaining, and preserving wild organisms for future study. People have been collecting natural history specimens for centuries and these specimens have been used for gathering valuable data about our changing planet (*Lane, 1996*; *Lister, 2011*; *Rocha et al., 2014*). Data collection from these organisms can include photographing, measuring, and obtaining blood or tissue samples for future molecular and/or biochemical analyses. These organisms are preserved in a fixative (such as formalin), dried, skinned, and stuffed, or otherwise prepared for long-term comparative use as a reference 'voucher' specimen (*Remsen, 1995*; *Buckner et al., 2021*; *Poo et al., 2022*). Vouchers are evidence of the existence of these organisms in a particular place and time that can be compared with specimens from other time periods and locations. These reference specimens document morphological, genetic, and phenological variation of a species or population and are used as proof of a species new to science (as type specimens). Scientific specimens can be used for countless research applications including as part of the 'extended specimen' concept (*Webster et al., 2017*; *Meineke et al., 2019*; *Lendemer et al., 2020*; *Monfils et al., 2022*). In particular, digitized collections facilitate large-scale studies on critical questions in global change biology (*Johnson, Owens & Global Collection Group, 2023*; *Ford et al., 2023*), and GBIF-enabled research (the most widely used specimen database is GBIF, the Global Biodiversity Information Facility) extends across all major science disciplines (*Heberling et al., 2021*). It should be clear that publicly available natural history collections facilitate opportunities for international and interdisciplinary collaboration (*Meineke et al., 2019*).

Unfortunately, natural history collections are in serious decline (*Rohwer, Rohwer & Dillman, 2022*). Part of this decline is the shifting in focus of traditional natural history museums away from collections research towards a focus on public exhibits and sustainability (*Naggs, 2022*), as well as cultural and political resistance to the idea of collecting certain organisms (*Byrne, 2023*; *Nachman et al., 2023*). It is also related to a decline in basic science research funding in natural history and the growth of more applied scientific research with economic goals (*Naggs, 2022*), leading to greater inequalities between researchers from resource-rich *versus* resource-poor countries or institutions (compare availability of research collections in the Global North *vs.* the Global South in the 'Institutions Holding Collections' S1 table from *Johnson, Owens & Global Collection Group (2023)*). These inequalities lead to the pressing need for institutions conducting collections-based research to connect and share opportunities more equitably. However, the future of taxonomic research in general will require funding for training, travel, and access to permits along with other infrastructure (*Britz, Hundsdörfer & Fritz, 2020*).

Here, we aim to highlight geographic regions where there are relatively few or no publicly available natural history records to raise awareness about regions that might be worthy of future natural history research and conservation. Future research may include, for example, aiding smaller, private, or non-digitized collections to become part of the global collections infrastructure by helping to make their vouchers and associated data accessible and public. Previous studies have used GBIF data to suggest areas to prioritize

for natural history work comparing voucher and observation records (*Daru & Rodriguez, 2023*) and to look at data gaps in patterns of digitization and sampling efforts (*Meyer et al., 2015*). Here we use voucher records with reliable locality information for terrestrial and freshwater habitats, and exclude the vast and still relatively poorly explored and enormous marine realm (*Costello et al., 2010*; *Albano et al., 2020*; *Molony et al., 2022*). We limited our searches to terrestrial and freshwater vascular plants, and to terrestrial vertebrates (*viz.*, birds, herpetofauna, and mammals) and freshwater fishes. Though oceans comprise the most abundant habitat on Earth, most natural history research has focused on terrestrial environments (*Dayton, 2003*; *Oestreich, Chapman & Crowder, 2020*). Therefore, the lack of collections in these terrestrial areas may be more notable and informative than that of the relatively overlooked and vast oceans. Additionally, we call for the disuse of the term 'unexplored' in natural history contexts which has been used both by academic and non-academic sources to mean an area previously uninvestigated or poorly known to Western Science (*Young, Petersen & Clary, 2005*; *Zou & Prasain, 2017*; *Du Chaillu, 1860*; *Smith, 2018*; *Schild, 2019*; *Montanari, 2017*). We aim to shed light on what could alternatively be meant by 'unexplored' by examining the different factors that explain the dearth of apparent collection activities in some parts of the globe. We suggest 'biodiversity blindspot' (defining it as "an area that may have a history of scientific collecting but that is lacking digitization records of natural history specimens") as a more accurate term to replace 'unexplored' in the context of natural history.

## MATERIALS AND METHODS

Occurrence records of preserved specimens with GPS coordinates were downloaded from GBIF (Global Biodiversity Information Facility; https://www.gbif.org/) for each of six taxonomic groups: fungi, terrestrial and freshwater vascular plants, freshwater fishes, herpetofauna (amphibians and reptiles), birds, and mammals (the retrieved records can be found at 10.6084/m9.figshare.26337067; and GBIF occurrence record downloads can be found for each group individually as Plants (*GBIF.org, 2022a*); Fungi (*GBIF.org, 2022b*); Freshwater Fishes (*GBIF.org, 2022c*); Birds (*GBIF.org, 2022d*): Mammals (*GBIF.org, 2022e*): Herps (*GBIF.org, 2022f*)). As we are interested in general trends in collection records across continents, only occurrence records with associated voucher specimens were considered for each of the six taxonomic groups included in this study and we did not include observational data where no specimens were collected. All data filtering and spatial analyses were performed in R version 4.3.1 (*R Core Team, 2023*). The GBIF records were filtered using the R 'CoordinateCleaner' package (version 3.0.1; (*Zizka et al., 2019*)), applying the 'capitals', 'centroids', 'equal', 'gbif', 'institutions', 'zeros', 'seas', 'duplicates', 'urban' tests. Please note: portions of this text were previously published as part of a preprint (*Ball et al., 2024*).

We identified both 'hotspots' and 'blindspots' of natural history collections using the Getis-Ord Gi* statistic, which tests for spatial autocorrelation (*i.e.*, the degree to which a set of spatial points is correlated to their geographic neighbors). A positive Gi* indicates clustering of high values (*i.e.*, high numbers of collections), while a negative Gi* indicates clustering of low values. The magnitude of Gi* indicates the strength of the clustering. A

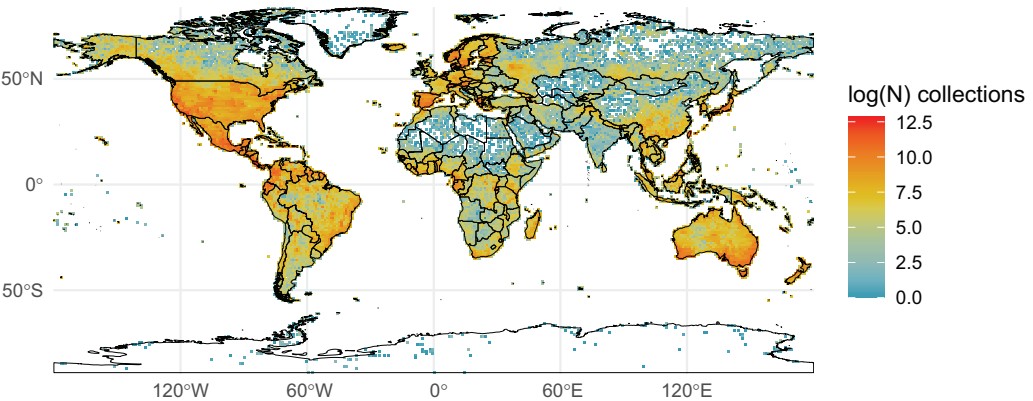

**Figure 1 Log-transformed Global Digitized Collections.** The number of global natural history collections per one-degree squared grid cell, log-transformed.

global spatial vector comprising the total number of collections per one-degree squared grid cell was created using the 'terra' R package (version 1.7–71) and used as input for the analysis. Getis-Ord Gi* was calculated in R using the 'local_g_perm' function from the 'sfdep' package (version 0.2.4) at default settings. In addition to calculating the Gi* value for each grid cell, 'local_g_perm' also performs a folded permutation test for the cell and outputs a p-value. The R script used to perform all spatial analyses as well as the spatial vector used as input can be found at 10.6084/m9.figshare.26360050.

An interactive map with unfiltered data (for comparison) was also generated using QGIS2Web plugin (ver. 3.16.0) and can be found at https://canon-network.github.io/ under the 'Specimen Database' tab. This map shows the total number of collected specimens divided into one-degree squared grid cells covering the globe and can be toggled for individual taxonomic groups.

We also examined the total number of preserved specimen records available on GBIF collected from each country compared to the number of preserved specimens in collections within the same country. We used the 'occ_count' function of the rgbif package in R to count the total number of preserved specimens collected in each country and count the total number of preserved specimens that were published by the same country in which they were collected. We additionally calculated the ratio between these figures. Specimen count data was retrieved on July 2, 2024. Issues related to GBIF and spatial bias and how they may not reflect presence or absence of taxa are well studied elsewhere (*Beck et al., 2014*).

## RESULTS AND DISCUSSION

We initially retrieved 55,200,291 natural history collection records across all taxonomic groups from GBIF. After filtering for duplicates and errors, we were left with a remainder of 40,932,450 total occurrence records for analysis. After filtering, terrestrial and freshwater plants represented most collections (90.7%), followed by fungi (3.0%), fishes (2.1%), herpetofauna (1.8%), birds (1.7%), and mammals (0.8%). A map showing the log-transformed numbers of filtered occurrence records per one-degree squared grid cells

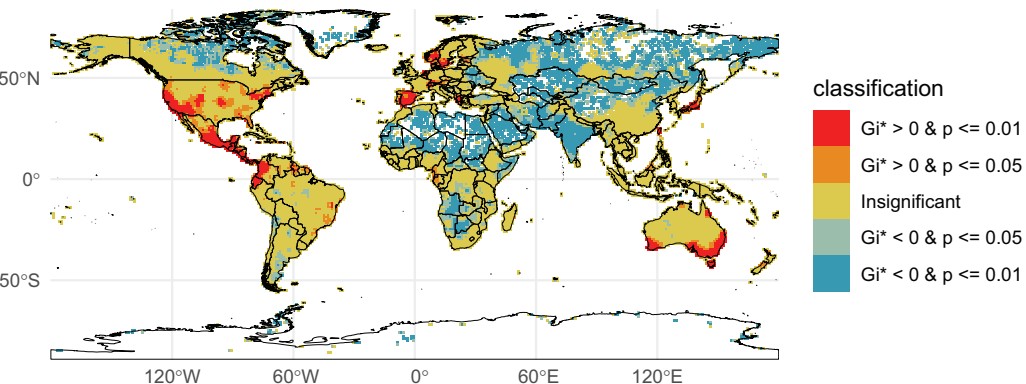

**Figure 2 Getis-Ord Gi* Global Collections.** Natural history collection hotspots and blindspots. Grid cells on the map are colored according to their corresponding Getis-Ord Gi* values and *p*-values (*p*-value of the folded permutation test). Red-colored cells indicate the most well-collected regions (hotspots) while dark blue and white cells indicate relatively poorly sampled regions (blindspots).

is shown in Fig. 1. The data was log-transformed to increase the color contrast of poorly sampled areas (*e.g.*, 'blindspots') so they would be more pronounced on the maps. The results of the Getis-Ord Gi* analysis (based on the raw numbers of occurrence records per one-degree squared grid cells) are shown in Fig. 2 and the remaining taxon based regional figures are also shown below (Figs. 3–8).

## South America

Publicly available collections in South America are more numerous in mountainous regions and well-known biodiversity hotspots (*i.e.*, the Andes, Atlantic Forest, and the Guiana Highlands; Fig. 3). This pattern is likely a result of collecting efforts being centered in well-known areas of high species richness that are relatively accessible. Similarly, digitized collections along the Andean mountains show a latitudinal gradient in sampling effort. This pattern is consistent across taxonomic groups and is likely linked to a well-documented latitudinal biodiversity gradient (*Fine, 2015*). Every group examined appears to have a high number of collections in the Atlantic Forest of Brazil and Uruguay relative to dryer adjacent regions including the savannas of the Cerrado in eastern Brazil. However, estimates of terrestrial vertebrate diversity made using publicly available data have identified hotspots like the Andean mountains as among the regions with the fewest voucher specimens (*Šmíd, 2022*). Assessing the extent of the scarcity of data collection in highly biodiverse regions remains a challenge. There are relevant records stored in private collections, as well as specimens at local institutions which have not, or have only partially been aggregated into digital repositories (*e.g.*, Museum of Zoology of the University of São Paulo, Cartagena Botanical Garden 'Guillermo Piñeres', the National Museum of Brazil, the Museo de Historia Natural UNMSM in Lima), which remain obscured to the international scientific community.

The Amazon Basin, one of the largest and most biodiverse regions of the world (holding 10% of all named plant and vertebrate species; (*Nelson et al., 1990*; *Schulman, Toivonen & Ruokolainen, 2007*; *Feeley, 2015*; *Winemiller et al., 2016*; *Stropp et al., 2020*; *Albert et al.,*

Peer J

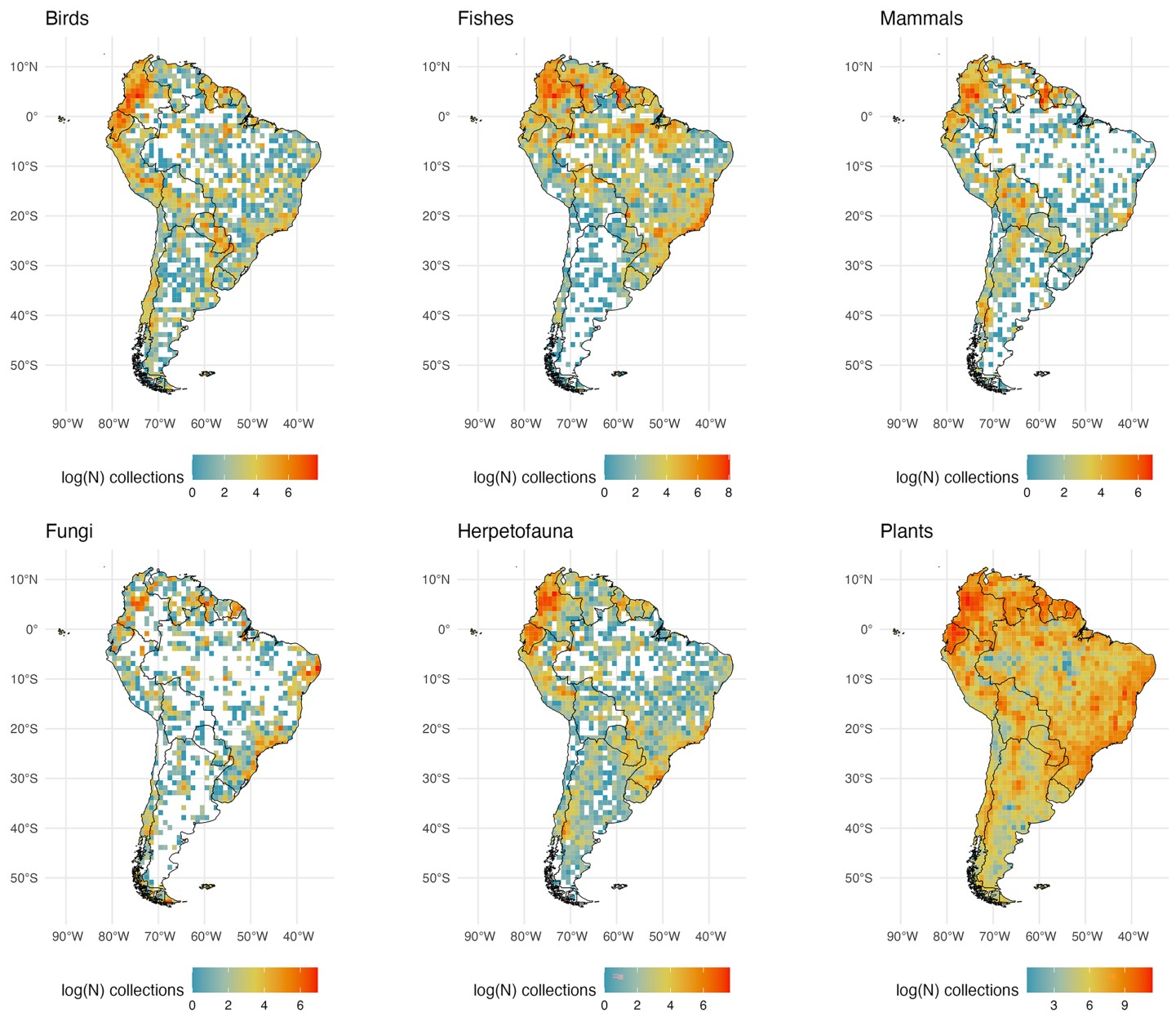

**Figure 3 South American Collections.** The number of global natural history collections per one-degree squared grid cell (log-transformed) in South America.

2023)), has lower sampling across taxa (Fig. 1). Large sections of the Amazon show very few plant samples, with botanical census data covering only a very small portion of the region (*Feeley, 2015*). Other lowland areas with fewer publicly available records across taxa include the Orinoco Basin, the Atacama Desert, the southern portion of the Chaco, and the Patagonian grasslands. While fewer digitized collections in the Atacama may be representative of the region's low diversity for the six taxonomic groups examined here, accessibility and armed conflict may be relevant factors limiting exploration in the other regions. Scarcity of roads or navigable water bodies are a challenge to transportation to and

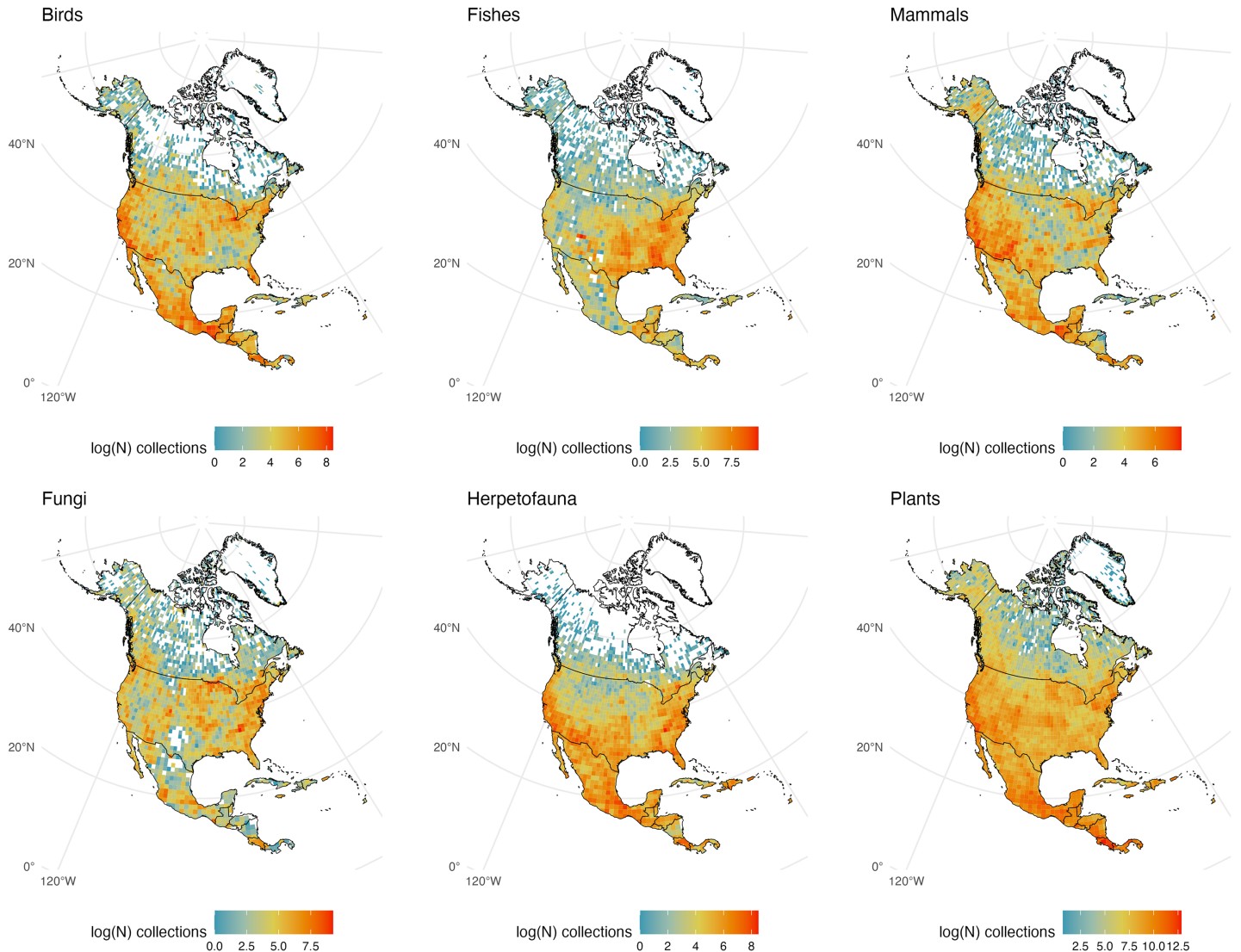

**Figure 4 North and Middle American Collections.** The number of global natural history collections per one-degree squared grid cell (log-transformed) in North America, Middle America, and the Caribbean.

from large sections of those regions, increasing the costs, risks, and logistics for field exploration (*Daru et al., 2018*; *Hijmans et al., 2000*). Political and civil unrest in Latin America have historically limited collecting in these and other regions and may have kept many collections there from being part of a global digitized database. This lack of digitization may be reflected in the dearth of collections represented in what are very biodiverse countries. In fact, such biodiversity is related in part to a positive relationship between forest cover and the intensity of armed conflict (*Álvarez, 2020*; *McNeely, 2003*; *Hanson et al., 2009*; *Negret et al., 2017*).

Specimen data for plants are the most abundant compared to the other taxonomic groups examined here. However, it has been estimated that there are zero publicly available botanical records from about 10% of tropical South America (*Feeley, 2015*). A higher

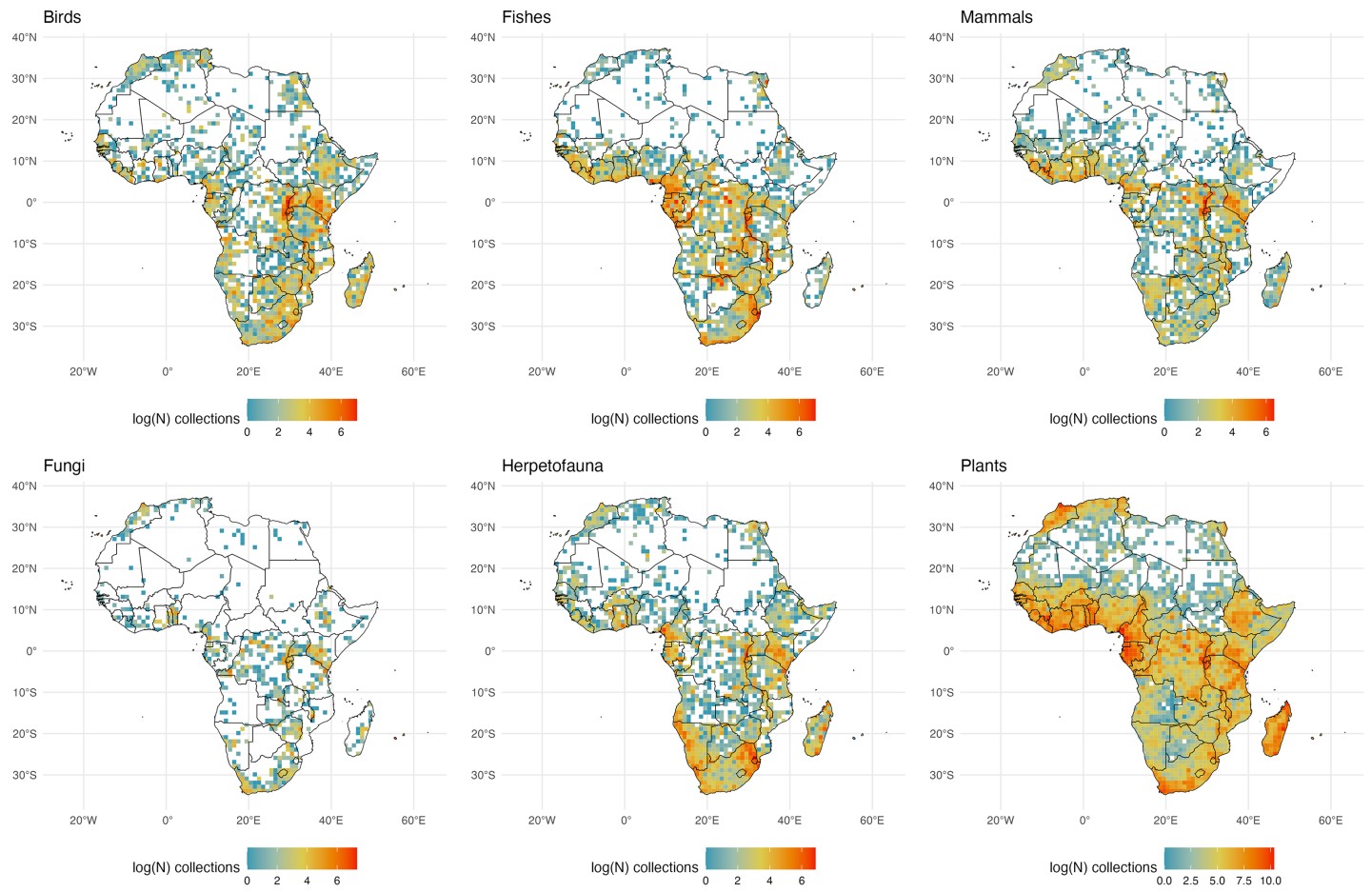

**Figure 5 African Collections.** The number of global natural history collections per one-degree squared grid cell (log-transformed) in Africa, Madagascar, and adjacent land masses.

number of botanical collections may be due to higher plant species richness compared to other groups.

Freshwater fishes are the most speciose group of vertebrates, and their species richness peaks in South America, specifically along the main tributaries of the Amazon River Basin (*circa* 7,000 species; (*Albert & Reis, 2011*)). The Amazon Basin and major tributaries appear much better collected (and digitized) than the Orinoco River Basin which covers much of Venezuela and Colombia. In Venezuela, only the Universidad Central has published data in GBIF (for insects), and many are privately held. The 'La Plata' region (including the Parana and Paraguay) in east central South America is also poorly sampled and/or digitized relative to the Amazon. Notably, the Deseado River estuary, Lake Musters, and Nuevo Gulf in Argentina are well-sampled for fishes. Compared to the other groups examined, there are comparatively fewer records of fungi. Intensively sampled regions for this group may be a consequence of the presence of institutions where this taxonomic group has been well studied and where collections have been digitized. For instance, southeastern Brazil appears more intensively sampled because the Herbarium at Federal

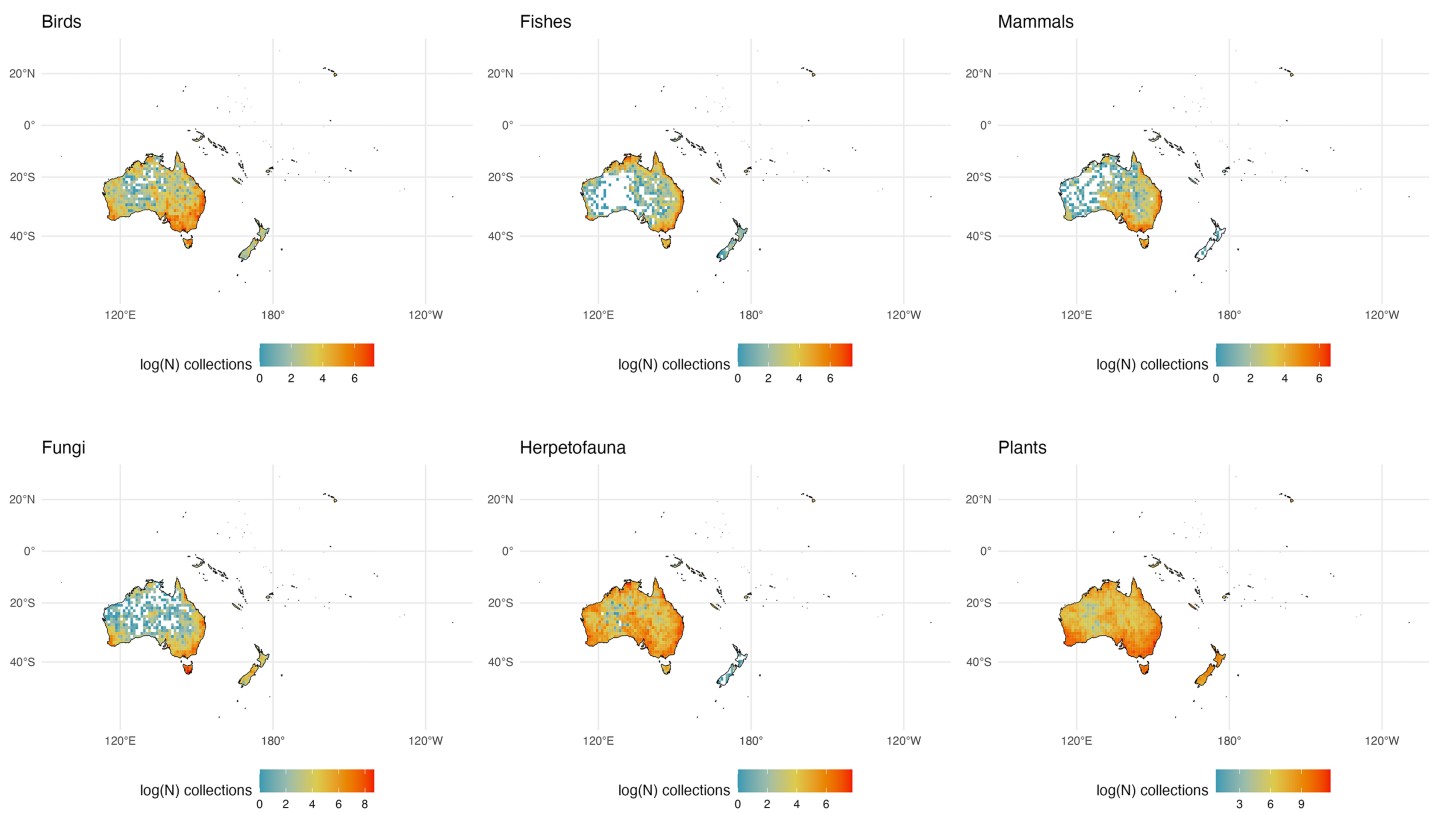

**Figure 6 Australia and Adjacent Islands Collections.** The number of global natural history collections per one-degree squared grid cell (log-transformed) in Australia and Adjacent Islands.

University of Pernambuco is located in this area and has the largest collection of fungi in Brazil (*GBIF, 2015*). Relative to northern South America, Brazil appears sparsely sampled for specimens of mammals, herps, and birds, especially relative to plants and fishes.

## North and Middle America (including landmasses in the Caribbean)

North America is one of the most consistently sampled continents with all taxonomic groups exhibiting similar decreases in sampling effort along a south-to-north latitudinal gradient and few areas devoid of specimen samples of any group south of Canada (Fig. 4). Excluding Greenland, interior Canada is the largest gap for all plant and animal groups examined, from Ontario to the Northwest Territories and east to Newfoundland and Labrador. Canada may show relatively few natural history samples in part because of its enormous size and climate (creating areas of remoteness); the lack of knowledge about its abundant freshwater resources has been previously noted (*Desforges et al., 2022*). Relative to adjacent countries it appears Nicaragua is poorly sampled for all vertebrate groups and fungi despite having an abundance of biodiversity-rich regions including the Mosquito Coast, Lake Nicaragua, and the largest tropical rainforest north of the Amazon (*Weaver, Lombardo & Martinez-Sanchez, 2003*). Otherwise, most of Central America and the Caribbean/Greater Antilles appear well sampled with a clear gap for Cuba, due more so to

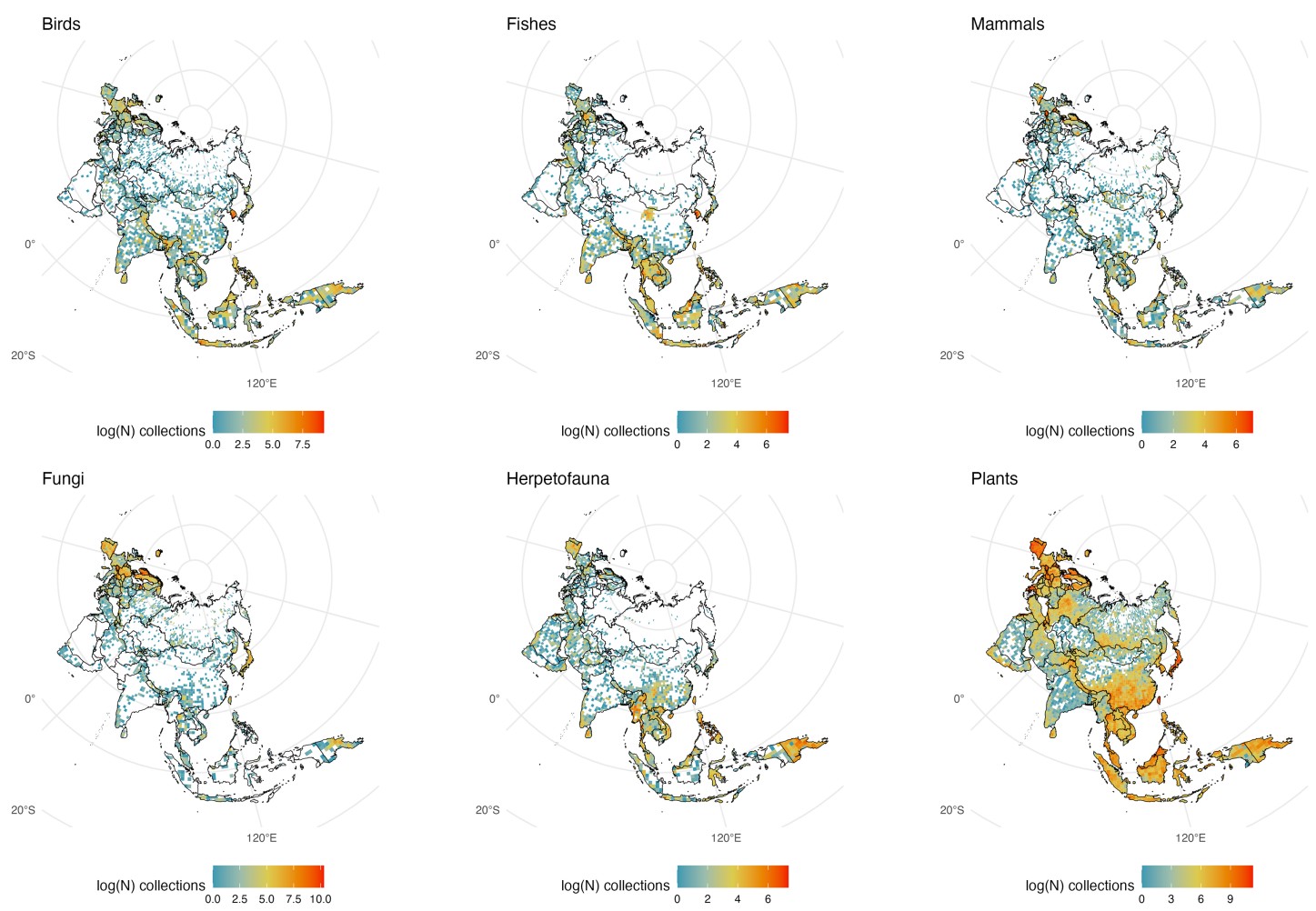

**Figure 7 Eurasia Collections.** The number of global natural history collections per one-degree squared grid cell (log-transformed) in Eurasia.

conflicting politics and policies (at least with the United States) than to a lack of biodiversity (*Denis, Cruz-Flores & Testé, 2018*). Conservation work in Cuba is among the best in the region (*Goulart et al., 2018*) and the Museo Nacional de Historia Natural de Cuba in Havana has digitized collections, but these are not publicly available. Perhaps not surprisingly the United States is very well sampled, and climatic differences between the wetter, more humid east *vs.* the more arid west can be seen when comparing fungi and fishes from those regions; with the more temperate Pacific Northwest being an exception.

Mammals have weaker collection records from the southeast and midwest United States than other vertebrate groups. Specimens of amphibians and reptiles (herpetofauna) display the sharpest decline moving northward of all examined groups, with few specimens collected in and east of the Rocky Mountains from Montana to Illinois (most likely do a drop in taxonomic diversity because of a climatic shift). Fungi and fishes are, unsurprisingly, sparsely collected from the northern Rocky Mountains south to central Mexico including adjacent desert regions. Although most vertebrate groups display strong

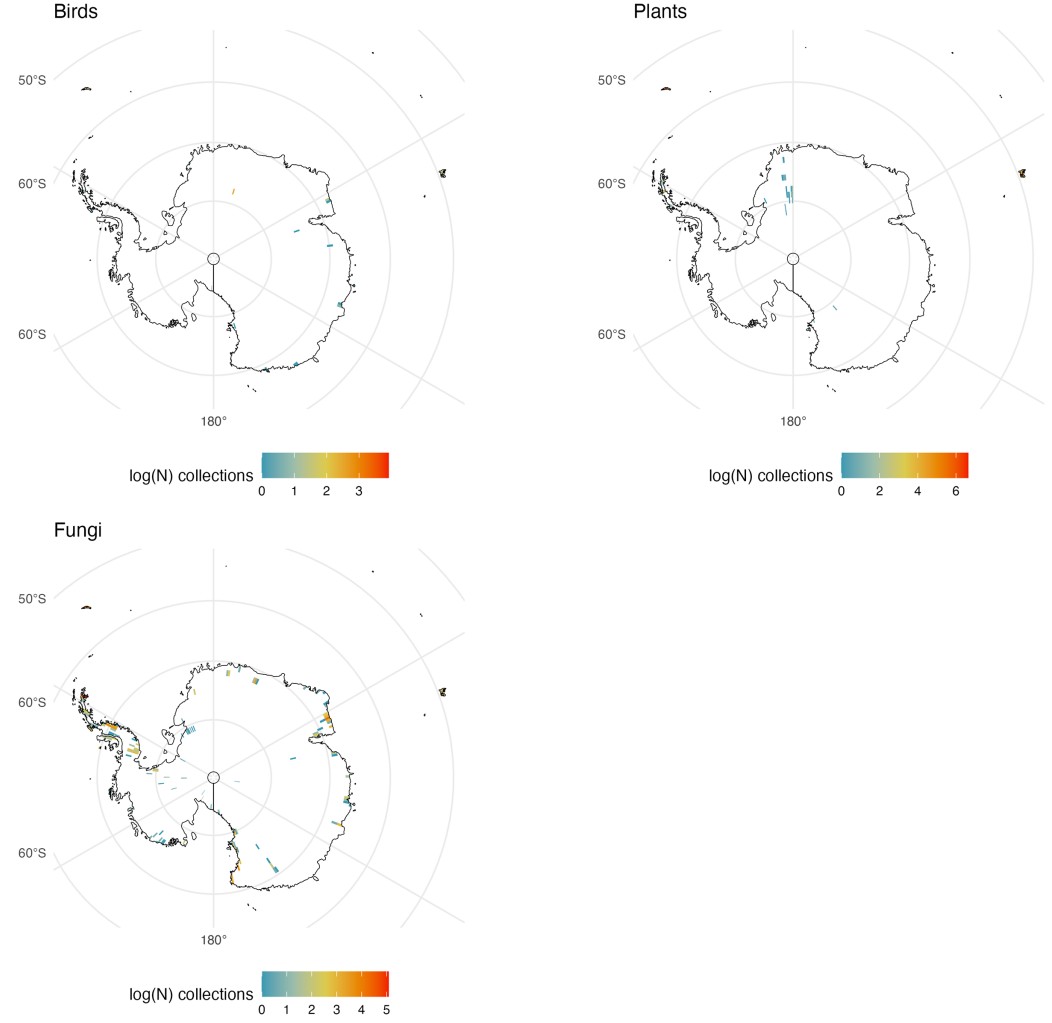

**Figure 8 Antarctic Collections.** The number of global natural history collections per one-degree squared grid cell (log-transformed) in Antarctica.

holdings from the southeastern United States, non-coastal southern Appalachia is relatively poorly represented in bird collections, as are Montana, Wyoming, and Utah in the west. Plants are the most consistently sampled group across North America with few, if any, regions that could be called a blindspot, while fungi have the highest number of areas lacking sampling according to GBIF, especially in and around the Chihuahuan Desert.

## Africa (including Madagascar and adjacent landmasses)

Perhaps more than any other region, Africa's 'biodiversity blindspot' areas likely reflect the sparsity of digitization of regional collections as well as the general lack of collections (Fig. 5). A wide variation in climate such as the dry Sahara region *vs*. sub-Saharan tropical regions, may be important explanations for variation in collecting effort within the continent, as these climatic factors are linked to both biodiversity and accessibility. For example, the remoteness of many places within the large Sahara Desert (a significant global

'biodiversity blindspot' across taxa; Fig. 2) create accessibility issues combined with low species richness for many taxonomic groups. Collections for most groups track with humidity and rainfall, save for fungal diversity which is likely higher than represented by the few collections that have been made across the continent, including Madagascar. Notably, Cape Verde is a bright spot for fungal collections on the West Coast of Africa almost entirely accounted for by collections from lichenologists at the Senckenberg Research Institute and Natural History Museum in Frankfurt, Germany, demonstrating the outsized impact a single research program can have on perceptions of whether an area has been studied (*Büdel & Mies, 1993*). A striking juxtaposition in known collections is seen between the relatively well-sampled island of Madagascar and adjacent islands such as the Seychelles. This dichotomy is particularly clear for plants and is likely a result of sampling efforts from the Missouri Botanical Garden which has a long history of botanical research on Madagascar: accounting for an estimated 72% of vascular plant collections from Madagascar. Unfortunately, this means the largest collection of Malagasy plants exists well outside of the island which would hinder the growth of local knowledge on these organisms (*Phillipson et al., 2006*). For mammals, the Great Rift Valley region is particularly well sampled, as are the southern reaches of West Africa (from Liberia to Benin). Madagascar has relatively few mammal specimens in collections despite its high proportion of endemic mammal species; this is likely due to the conservation status of many of the species, including the endangered endemic lemurs.

Despite being largely tropical, Somalia, the Central African Republic, and large areas in the Democratic Republic of the Congo, Angola and Botswana are significant 'biodiversity blindspots' across taxonomic groups (Fig. 2). The sparsity of collections in these regions can be attributed to a combination of complex factors including a history of colonization, lack of scientific infrastructure and institutional support, geographic accessibility, and historical and present-day political instability (particularly in central Africa where plant collections have been noted to decline during periods of war (*Sosef et al., 2017*)). While not exclusive to these poorly collected regions, cultural factors are also an important consideration (*e.g.*, in some areas there may be cultural beliefs that discourage the collection of natural history specimens (*Stropp et al., 2020*; *Sosef et al., 2017*)). One additional factor is that the Royal Museum for Central Africa (Belgium) reports 10 million biological specimens in its collections from the Congo and adjacent regions, but only a small percentage (<0.5 million) of these are on GBIF (E. Greenbaum, 2024, personal communication).

We call for collections in Africa to join the GBIF network but also call on institutions outside of Africa with African collections to publish their records (some Western museums may have collections from the region that are not yet digitized–as noted above). Other countries, such as the Central African Republic, have no digitized collections, and all the digitized voucher records from that country are housed in other countries (Table 1). Egyptian collections from Cairo, Alexandria, and along the Red Sea for several groups are notable and likely were a target for several important historical collections. Interior reaches of Africa, outside of the desert regions, including the Okavango Delta still have relatively few collections digitized (*Tolley et al., 2016*; *Greenbaum, 2017*). Regional museums such as

**Table 1 GBIF records by country.** The total number of preserved specimen records available on GBIF (Global Biodiversity Information Facility) collected from each country represented in the GBIF network (gbif.org), compared to the number of preserved specimens in collections in the same country. These data show that for many countries the majority of specimens of organisms from that country are housed outside of those countries. Note that some counties shown here may have their own collections (even large ones such as the Smithsonian Tropical Research Institute collections from Panama), but they are either not digitized with GPS locality information or are not on GBIF, or both.

| Country | Total GBIF specimens | In-country GBIF specimens | Ratio |
|---|---|---|---|
| Andorra | 17,016 | 2,084 | 0.12 |
| Angola | 252,941 | 34,035 | 0.13 |
| Argentina | 1,567,321 | 762,734 | 0.49 |
| Armenia | 109,370 | 22,332 | 0.20 |
| Australia | 13,531,876 | 11,920,608 | 0.88 |
| Belarus | 26,762 | 434 | 0.02 |
| Belgium | 1,573,733 | 1,454,636 | 0.92 |
| Benin | 106,287 | 19,134 | 0.18 |
| Brazil | 14,809,662 | 11,804,968 | 0.80 |
| Cambodia | 44,726 | 0 | 0.00 |
| Cameroon | 579,737 | 24,637 | 0.04 |
| Canada | 9,424,480 | 5,747,819 | 0.61 |
| Central African Republic | 87,827 | 0 | 0.00 |
| Chile | 752,207 | 120,543 | 0.16 |
| Colombia | 3,311,308 | 2,051,920 | 0.62 |
| Costa Rica | 5,580,581 | 3,948,851 | 0.71 |
| Croatia | 131,772 | 494 | 0.00 |
| Denmark | 1,329,807 | 978,502 | 0.74 |
| Ecuador | 1,908,812 | 137,806 | 0.07 |
| Estonia | 1,193,114 | 1,162,799 | 0.97 |
| Finland | 3,779,722 | 3,443,474 | 0.91 |
| France | 2,589,044 | 949,048 | 0.37 |
| Georgia | 105,787 | 8,400 | 0.08 |
| Germany | 2,509,530 | 1,496,017 | 0.60 |
| Guatemala | 577,865 | 47,177 | 0.08 |
| Guinea | 125,872 | 11,069 | 0.09 |
| Iceland | 386,870 | 150,935 | 0.39 |
| Ireland | 403,781 | 9,679 | 0.02 |
| Kenya | 549,942 | 70,844 | 0.13 |
| Liberia | 105,456 | 0 | 0.00 |
| Luxembourg | 58,221 | 33,102 | 0.57 |
| Madagascar | 1,527,643 | 60,390 | 0.04 |
| Malawi | 173,249 | 21,567 | 0.12 |
| Mauritania | 34,275 | 1,752 | 0.05 |
| Mexico | 11,087,717 | 6,370,268 | 0.57 |
| Namibia | 314,200 | 9,732 | 0.03 |
| Netherlands | 2,212,294 | 1,916,920 | 0.87 |

(Continued)

| Table 1 (continued) | | | |
|---|---|---|---|
| Country | Total GBIF specimens | In-country GBIF specimens | Ratio |
| New Zealand | 1,951,215 | 1,479,535 | 0.76 |
| Nigeria | 211,582 | 45,214 | 0.21 |
| Norway | 5,254,894 | 4604827 | 0.88 |
| Panama | 1,045,272 | 0 | 0.00 |
| Peru | 1,836,957 | 14,077 | 0.01 |
| Poland | 3,364,339 | 3,189,123 | 0.95 |
| Portugal | 791,878 | 376,188 | 0.48 |
| Sierra Leone | 71,796 | 0 | 0.00 |
| Slovakia | 224,297 | 118,869 | 0.53 |
| Slovenia | 64,170 | 14,809 | 0.23 |
| South Africa | 3,261,325 | 2,215,243 | 0.68 |
| South Korea | 3,178,124 | 3,072,768 | 0.97 |
| Spain | 47,85,853 | 3,924,075 | 0.82 |
| Sudan | 70,992 | 0 | 0.00 |
| Suriname | 321,064 | 13,173 | 0.04 |
| Sweden | 5,760,715 | 5,177,353 | 0.90 |
| Switzerland | 31,08,495 | 2,712,250 | 0.87 |
| Tajikistan | 57,360 | 9,176 | 0.16 |
| Tanzania | 772,156 | 33,838 | 0.04 |
| Timore-Leste | 16,445 | 0 | 0.00 |
| Togo | 46,941 | 12,422 | 0.26 |
| Tonga | 44,122 | 0 | 0.00 |
| United Kingdom | 3,049,482 | 2,634,811 | 0.86 |
| United States | 48,944,170 | 46,496,186 | 0.95 |
| Uruguay | 106,725 | 8,130 | 0.08 |
| Uzbekistan | 52,617 | 10,839 | 0.21 |
| Zimbabwe | 248,281 | 56,575 | 0.23 |

the Port Elizabeth Museum (South Africa) that have known collections from the Okavango and surrounding regions are not currently linked to GBIF. Given that region's celebrated conservation status, efforts to collect in this area should be well-regulated. Notably many collections from this region are in Europe and the United States, and efforts to digitize existing collections and create new regional collections should be considered as they would benefit local knowledge.

## Australia and Adjacent Islands

Unsurprisingly, the large swath of arid land through central Australia is less densely sampled than tropical northern regions and the temperate south, due to the relative dearth of species biodiversity in this arid climate, particularly for large vertebrates (Fig. 6; *Dickman, 2018*). This portion also remains logistically difficult to access for collecting, due to its desert climate and relative lack of travel infrastructure (*i.e.*, roads). Tasmania is a

significant global hotspot for specimens across taxonomic groups, as is the region near Perth in the southwest, due in part to their more tropical climates, easier accessibility, and the presence of natural history museums in each locale (*Dickman, 2018*; Fig. 2). The lack of mammal and herp collections from New Zealand is not surprising because of the relatively few species of these groups known to exist there.

## Eurasia

Much of Europe has been well collected and the number of digitized specimens from the region is among the highest on the planet; in stark contrast to the rest of Eurasia except the far East (particularly Japan and Taiwan which have substantial collections and active natural history researchers; Fig. 7). The relative absence of digitized collections from India for plants, fungi and mammals is unexpected, given the region's rich biodiversity. Sri Lanka appears well-sampled relative to India for several groups which may reflect a greater accessibility to collections and digitization of specimens from that island nation. South Korea, Nepal, and Taiwan appear to have a high number of collections of birds, mammals, plants, and herps relative to much of 'mainland' China, although the warm temperate areas of Southern China appear well collected for some groups such as plants and herps making the absence of mammals and birds from this region even more striking. Much of the cold remote vastness of Russia lacks GBIF representation for natural history specimens. Yet bright spots on the Russian map shed light on the importance of local museums in the pursuit of cataloging biological diversity. For example, the high concentration of fungi samples near the western central part of the country is not a centroid artifact but the Yugra State University Biological Collection: an institution that has been collecting mycological specimens since the early 20th century (*Filippova et al., 2020*). Lack of digitization, collecting efforts, climate, and politics probably have a great deal to do with the lack of vouchered samples from the world's largest country.

For some taxa, a sharp contrast exists in New Guinea between New Guinea (the western Indonesian side often called 'Irian Jaya') and the Eastern side of the island, Papua New Guinea, which has been a target for many natural history expeditions (*Mayr & Gilliard*; *Webb, 1995*; *Cookson, 2000*). Another noticeable contrast occurs between north and south of the New Guinea Highlands, with the south being poorly collected *versus* its northern counterpart. This is the case for both the Indonesian and Papuan halves of New Guinea, most likely due to the heavy monsoon flooding that seasonally submerges areas in that portion of the island (*Tanaka, 1994*).

Despite its tropical climate and abundance of biodiversity, there appear to be few collections from Indonesia, particularly for fungi. Contrastingly, the Philippines are densely collected relative to Indonesia, again showing how Indonesia appears poorly collected for natural history specimens. It is also possible that Indonesia only appears as a 'biodiversity blindspot' because of a dearth of accessible digitized specimens (likely in collections in the West). Indonesia has been a target for museum-based collections since occupation by the Dutch and Alfred Russel Wallace's early expeditions to the Indo-Australian Archipelago. Few if any of the specimens from southeast Asia collected by Wallace or used as type material by Western zoologists are curated by local institutions.

Specimens collected from this region more recently are curated in large institutions like the Museum Zoologicum Bogoriense in Java, Indonesia, which has yet to have its extensive collections digitized. The absence of a fungarium (despite there being herbaria) is an oversight and can be linked to why there are few fungal collections in an area that should be a hotspot for mycologists. Borneo appears remarkably underrepresented in natural history collections for herps and mammals, particularly relative to nearby Java.

Mammals and herps are particularly well collected in the Philippines and Papua New Guinea making adjacent areas even more glaring for their lack of representative samples on GBIF. Birds and plants appear to be the best sampled taxa from Indonesia, perhaps as a result of historic collecting efforts but gaps remain in New Guinea, Borneo, Sulawesi, and Sumatra.

Notably, no museum collection from Oceania (the scattered islands of the Pacific) is listed among the 73 of the world's largest natural history museums and herbaria from 28 countries, with the nearest of these surveyed collections being in India and Australia (*Johnson, Owens & Global Collection Group, 2023*). Representation of people from this region and digitization of local collections should be prioritized to further our understanding of the biodiversity of the region.

### Antarctica

Antarctica is notable only in the absence of many collections for obvious reasons. There are no collections of freshwater fishes, amphibians or reptiles because none survive on the continent. Large mammal and bird collections do exist but are generally restricted to historic collections or from areas near the coasts including some not yet recorded on GBIF. Similarly, fungi and plants are generally restricted to ice-free areas of the continent (Fig. 8). Observations of Antarctic fauna have been made but many of these are of marine invertebrates and fishes below the sea ice (*Chakrabarty et al., 2021*).

## CONCLUSIONS

The term 'unexplored' can be used in many unrelated contexts, and it may not accurately reflect the state of knowledge in a given area. To some, 'unexplored' may mean an area that has never been seen by human beings, while to others, it may mean an area that has not been studied by Westerners or is uninhabited. In examining the so-called 'unexplored' places that remain on land, a lack of digitized natural history collections may be a common cause for this label. In some cases, there may be a lack of natural history samples from an area because it is remote and there are relatively few organisms there to collect (*e.g.*, the Atacama Desert in Chile and Sahara Desert of Africa which are some of the driest places on Earth, as well as many parts of Antarctica). In some regions, political unrest has prevented access to naturalists (*e.g.*, Libya, Venezuela), and a lack of infrastructure for obtaining permits and permissions hurt both the global understanding of a region and local biodiversity education. Other areas appear simply to have been overlooked by natural history research—the limited number of practicing naturalists with even more limited funding may be an explanation for some of these regions remaining poorly researched. The term 'unexplored' in these contexts can be harmful, as such terminology may inadvertently

perpetuate misconceptions or undermine the value of existing research and the efforts of local scientists and indigenous people or disregard the underlying resource-imbalances impeding natural history research, collection, and digitization in many regions (*Ramírez-Castañeda et al., 2022*). For these reasons, we advocate for replacing the word 'unexplored' in natural history contexts for more precise and inclusive language (*e.g.*, 'this region lacks digitized natural history collections') and suggest 'biodiversity blindspots' as an alternative.

In this article, we highlight areas lacking or having few collections, not to encourage their exploitation, but to call for an increased understanding of these areas in the context of global biological diversity for the sake of conservation and recognition in a natural history context. A multitude of factors have kept these places from the growing knowledge base of natural history and biodiversity, and in some cases, particularly where indigenous stakeholders are the protectors (*Fletcher et al., 2021*), traditional Western approaches to science may not be the most effective means of including these regions (*Demery & Pipkin, 2021*; *Hernandez, 2022*). Good faith partnerships and collaborations, a fair exchange of knowledge and resources, and the acknowledgement that there are many different ways of knowing that are equally meaningful, should be necessary starting points to conversations related to natural history in many of these areas (*e.g.*, large swaths of the Amazon Rainforest that are home to uncontacted tribes). Recent calls for equity-based fieldwork should be heeded, and special care should be taken to avoid practicing parachute science, a term that refers to situations in which researchers, often from more economically developed or privileged countries, visit other marginalized countries to conduct research and then return to their home institutions without genuinely engaging or collaborating with locals (*Fletcher et al., 2021*; *Demery & Pipkin, 2021*; *Hernandez, 2022*; *Paknia, Rajaei & Koch, 2015*; *Culotta, Chakradhar & Pérez Ortega, 2024*). Some of the most biologically diverse and well collected countries have high incidences of parachute science. Any future work in regions lacking natural history samples should be performed in ways that are considerate of and inclusive to local researchers and communities. Likewise, the creation of local natural history collections will create a better understanding of biodiversity in the region and aid local and foreign researchers in the future.

Accessibility to financial resources that make digitization (especially large-scale efforts) possible are a significant barrier for many museums and institution—even for some larger institutions in the West (*e.g.*, based on GBIF data most natural history specimen collections from Costa Rica and Malawi, like many other nations, are found outside of those countries; Table 1). There are large museums that are not yet digitized and part of GBIF in some of the most biodiverse places, including but not limited to the Museum of Zoology of the University of São Paulo, the Museu Paraense Emílio Goeldi, Inpa Coleçoes Zoológicas, and the National Museum in Rio de Janeiro (which suffered a devastating fire in 2018; (*Lucia Araujo, 2019*)). There are also large museums in the West that appear to have a relatively small percentage (<15%) of their total collections on GBIF (*e.g.*, the Smithsonian National Museum of Natural History, National History Museum (London), Museum national d'Histoire naturelle (Paris); Museum für Naturkunde Berlin; these and others can be found under https://scientific-collections.gbif.org/institution/search) although we caution that "total collections" may include fossils, invertebrates, and other

"objects" not related to the vertebrate and plant specimens we are discussing here. Modernizing access to the reference vouchers housed at these institutions would likely benefit research groups globally and facilitate natural history and conservation research. However, international digitization efforts will require sharing resources to create scientific infrastructure where it is lacking, especially in places where biodiversity is richest and the people poorest. The benefits of digitization will not be realized equitably without global collaborative efforts that are inclusive to regions with fewer economic resources, and offer a fair exchange of knowledge, technology, and support to empower these areas in their scientific endeavors.

The goal of this study was to increase our knowledge of 'biodiversity blindspots' by highlighting these poorly-collected or poorly-digitized places, and we hope these focus areas can be topics of discussion by researchers, politicians, indigenous stakeholders, and others who are interested in the discovery-based science of biodiversity research. Ultimately, we call for the digitization of all collections, prioritizing those from 'biodiversity blindspots', and for their integration into global databases like GBIF. Our work may help recent advances in predictive modeling that can be used to track regions with a high potential to hold hidden biodiversity (*Parsons et al., 2022*). We hope this study will help identify the gaps in our understanding of the world's biodiversity and draw attention to how we identify areas that have been potentially overlooked by those interested in the exploration and conservation of the natural world.

## ACKNOWLEDGEMENTS

Ana Bedoya, Greg Thom, and Nathan Lord helped at various stages of manuscript preparations. We thank the thousands of natural history researchers, explorers, and collectors who contributed to the samples we examined here. We also thank the hundreds of museum researchers who prepared, digitized, and curated these voucher samples. We are also grateful to the databasing experts, particularly those at the Global Biodiversity Information Facility who have made this type of information widely accessible. We also thank two anonymous reviewers, and Eli Greenbaum and Editor Stuart Pimm for their comments on our work.

### Funding

This work was supported by the George H. Lowery Professorship and E.K. Hunter Chair to Prosanta Chakrabarty. The funders had no role in study design, data collection and analysis, decision to publish, or preparation of the manuscript.

### Grant Disclosures

The following grant information was disclosed by the authors:
George H. Lowery Professorship and E.K. Hunter Chair to Prosanta Chakrabarty.

## Competing Interests

The authors declare that they have no competing interests.

## Author Contributions

- Laymon Ball conceived and designed the experiments, performed the experiments, analyzed the data, prepared figures and/or tables, authored or reviewed drafts of the article, and approved the final draft.
- Sheila Rodríguez-Machado performed the experiments, prepared figures and/or tables, and approved the final draft.
- Diego Paredes-Burneo performed the experiments, prepared figures and/or tables, and approved the final draft.
- Samantha Rutledge performed the experiments, prepared figures and/or tables, and approved the final draft.
- David A. Boyd performed the experiments, prepared figures and/or tables, and approved the final draft.
- David Vander Pluym performed the experiments, prepared figures and/or tables, and approved the final draft.
- Spenser Babb-Biernacki performed the experiments, prepared figures and/or tables, and approved the final draft.
- Austin S. Chipps performed the experiments, prepared figures and/or tables, and approved the final draft.
- Rafet Ç. Öztürk performed the experiments, prepared figures and/or tables, and approved the final draft.
- Yahya Terzi performed the experiments, analyzed the data, prepared figures and/or tables, and approved the final draft.
- Prosanta Chakrabarty conceived and designed the experiments, performed the experiments, analyzed the data, authored or reviewed drafts of the article, and approved the final draft.

## Data Availability

The data is available at figshare: Ball, Laymon (2024). GBIF records. figshare. Dataset. https://doi.org/10.6084/m9.figshare.26337067.v1.

The R script is available at figshare: Ball, Laymon (2024). Spatial analyses. figshare. Dataset. https://doi.org/10.6084/m9.figshare.26360050.v1.

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
