# Peer review of "What ‘unexplored’ means: mapping regions with digitized natural history records to look for ‘biodiversity blindspots’"

_PeerJ, doi:10.7717/peerj.18511_

## Round 0.1 · original submission · Major Revisions

As you can see, all three of my reviewers recommend a major revision. I think you are most fortunate in that these reviewers have all made detailed, specific recommendations. I hope you will be able to address all of these reviews and return a corrected manuscript, along with a document answering all of these comments to us at your earliest possible opportunity.

Reviewer 1 ·

Basic reporting

I appreciate that the manuscript is well-written and clearly guides the process and findings. However, some information needs to be included that would increase its value.
First, the title suggests the paper will provide conceptual clarity from the beginning, but the authors discuss retiring the term “unexplored” only in the conclusions section.
The authors may well include in their work two essential studies given their similarities in approaches. Daru and Rodriguez (2023) test whether voucher collections represent current biodiversity distribution patterns better than non-voucher observations. They provide maps of concentrated voucher collections at different resolutions. This study is beneficial in its methods and findings. The map they produce can be used to estimate priority exploration areas for natural history specimens.
Another study (Meyer et al., 2015) uses biodiversity records regardless of type of observation to estimate gaps in digitized information and offer robust evidence on the factors determining distribution patterns of sampling efforts.
The authors will need to cite this useful information and make it clear how their work differs from previous findings.

Experimental design

I like that the analysis is simple, but it must be fully convincing, given the potential bias in the data. The authors may again follow Daru and Rodriguez (2023) as an example of data depuration. Using an enormous amount of data from GBIF, Daru and Rodriguez (2023) pay attention to removing duplicates that may overestimate sampling effort. Although I understand the authors of the submitted manuscript may want to show areas that have been oversampled, their methods do not allow to discriminate areas where many collections belong to a few taxonomical units or copy vouchers of a single species and areas with few collections but more diverse. This is critical for sampling prioritization and defining the so-called blind spots.
Other severe concerns regarding data validity are the potential of misplaced records and coordinate uncertainty. GBIF, although highly useful, contains records with wrong coordinates that may be associated with collection age. Older records are more prone to be misplaced due to higher geospatial inaccuracy. Other data coordinates might have been regarded as the final physical storage place rather than the recording location. See Table 1 in Stropp et al. (2020) for further reference on the filtering process.
The mapping of unexplored regions tells where it has been chiefly explored but not where blind spots are. Assuming blind spots are areas that may contain collections but have not been formally digitized or vouchered, this is better done by calculating the proportion of voucher collections relative to the entire records (with and without vouchers) per unit area. This method reduces overestimations of unsampled areas and allows for the setting apart of areas that are not only unexplored but also poorly sampled.

Validity of the findings

Given the lack of constraints in the filtering process of raw data and the estimation of unexplored areas based solely on the number of collections per unit area, the findings may be highly inaccurate. A thorough analysis must conceive the nature and origin of raw data and reduce bias due to geospatial uncertainty. I consider the work’s approach original but lacks rigor in design and methods.

Additional comments

Further comments:
L27 Abstract: Provide more details on the research context. Briefly state the problem and provide the main findings.
L50 Blind spot: Add definition here. It seems that it logically includes specimens that have not been digitized (L46-48), but in the way it is presented here, it appears as something separated.
L52-54 First goal: “Unexplored” and L81-83 second goal: Mapping global undersampled regions. Conflicts arise regarding the primary purpose of the paper and the main goal. I understand the authors want to map regions that differ in their sampling efforts regarding natural history collections. That would be the primary purpose; from there, you discuss concepts. Otherwise, you would require more than a mapping approach to defining “unexplored.” See Meyer et al. (2015) in how they provide different aspects to define the lack of sampling efforts. In the same line, authors use “unexplored” and “undersampled” or “uncollected” interchangeably. However, these are different concepts. Unexplored refers to where no efforts at all have been made. Undersampled or underexplored is where minimal efforts have been made.
L93 Meyer
L96-98 Blind spots definition. How does this differ from unexplored areas?
L103-104. Provide methods for identifying protection gaps.
L110 Methods: GBIF. The biases inherent in this kind of data are well known. See, for example, Beck et al. (2014)
L124-127 If the data cleaning processes included only removing noncontinental records, data is subject to high uncertainty. Duplicates, for example, may show exceptionally well-sampled areas while others are not. Another significant bias is regarding wrong localities. Coordinates of historically collected data, particularly museum specimens’ data, tend to be associated with the final physical repository rather than the original collection site. A careful review of identifying and filtering out these records would positively impact the results. Check the “Issues” and “CooridnateUncertaintyInMeters” columns in the dataset for each taxonomic group.
Biases might also be associated with taxonomic groups. For birds, for example, GBIF contains data from the eBird platform, which offers filtering options for historically and recently collected data, such as completeness and verification of checklists. Non-eBird records might be filtered based on their spatial relation to depurated eBird records.
L121 R version?
L140-143 Add details about climate factors in the Introduction and Methods. Why would temperature influence the collection rate? In L45, authors mention deserts as an example of poorly explored areas, but no evidence shows this relationship. Deserts have a more meaningful variable that may affect collections, such as remoteness. Besides using temperature as a factor of data sampling, I suggest using remoteness. This may make things more complex. In such a case, I would only refer to the mapping, and the discussion would offer some well-supported assumptions on the factors influencing differences in data collection.
L160-163 Thanks for providing the links for the raw data. Plants data is not available for download.
L164-166 This is related to my previous comment in L124-127. The issue here is about more than locality names but mainly coordinates designation.
L167-168 Besides temperature, the authors need to add more detail about how sampling may be biased toward sites with research or museum institutions. Read Meyer et al. (2015), for instance.
L189-191 This sentence conflicts with that in L173-174. Thus, it is not a pattern that well-known species-rich areas are well sampled. It is geography dependent.
L192 Is this a value calculated from the data of it is from the reference you cite. If the former, how was it obtained? I did not see it in the reference.
L237-238 I do not think size matters when you compare Canada with the US. Remoteness and less appealing areas for species richness may play a more critical role in Canada. See Meyer et al. (2015) for other similar factors.
L264-266 How the weather may influence the sampling effort must be clarified. Provide supporting evidence. I think the Sahara region is a matter of remoteness or accessibility and low richness.
L271 Cite the assertion.
L337-339 What is the basis to state that this could be labeled? Has someone already attempted to provide such connotation to that area?
L379-381 It must be clarified if the manuscript intends to solve a conceptual issue or provide evidence of regions’ missing sampling effort in natural history collections. All this paragraph and the idea of defining terms must be restructured since the conceptual issue cannot be solved simply by looking at a map. Ideally, a review paper may address how the term “unexplored” is used in biodiversity studies and propose the blind spots for natural history collections. These blind spots, however, must be based on where they have already been sampled with no vouchers to reflect that records exist but are unvouchered.

Suggested references
Daru, B.H., Rodriguez, J. (2023). Mass production of unvouchered records fails to represent global biodiversity patterns. Nat Ecol Evol 7, 816–831. https://doi.org/10.1038/s41559-023-02047-3
Meyer, C., Kreft, H., Guralnick, R. et al. (2015). Global priorities for an effective information basis of biodiversity distributions. Nat Commun 6, 8221. https://doi.org/10.1038/ncomms9221
Stropp, J., Umbelino, B., Correia, R.A., Campos-Silva, J.V., Ladle, R.J. and Malhado, A.C.M. (2020), The ghosts of forests past and future: deforestation and botanical sampling in the Brazilian Amazon. Ecography, 43: 979-989. https://doi.org/10.1111/ecog.05026

·

Basic reporting

Review of Unexplored Regions Paper

Based on the title, abstract, and conclusions, the main point of this paper is to address what is meant by the term “unexplored.” An interesting premise, but who, exactly, is promulgating this term in the literature? The first time the term is noted is in quotation marks on line 53 of the Intro, but no citation is provided. It is mentioned once more in the Intro on line 99 as a historical term, but again no citation is provided. To ensure the main point of the paper is addressed, I believe the authors must first document how this term has been used, both in terms of biodiversity science and perhaps exploration too, in both historical and modern contexts. If they do this with a thorough interdisciplinary literature review, I believe the focus of this paper will be clear and of interest to a wider audience.

The other issue I have with this paper is about the main conclusion. It seems to me (and the authors reiterate this in places) that there are two possibilities to explain the sampling gaps identified in the paper: 1) very few people (or none at all) have collected in the area in question, or 2) there are collections from the area in question, but they are not digitized and populated into GBIF, the only source of data for this study. It would be super helpful to know how many collections are missing from the current GBIF database, and where the foci of those collections are. I realize this is a huge ask that will likely require a lot of time, but without this, it seems that the authors don’t really know whether the gaps are primarily due to 1) or 2) as noted above. One possible way of attacking this would be to reconcile the list of collections in Sabaj et al. (ASIH list of acronyms) with the list provided on GBIF- the ones that are missing could then be queried to see what their taxonomic and geographic strengths are.
I hope these comments are helpful for a revision- the overall premise of the study is sound and I hope it can be published down the road. Below are a few minor issues:
Line 43: I see that Tsingy refers to a national park in Madagascar with jagged limestone formations, but it is not a word in common usage, and the authors should replace it with something more recognizable
Line 275: I got lost here, because there was no “diagonal band” pattern noted for plants in the previous paragraph that could be reconciled with the pattern for herps- I looked at the maps and I could not see any obvious similarities to explain this.
Line 289: given that my research focus has been on DR Congo, I would emphasize that the perceived lack of collections in this area is due to the Royal Museum for Central Africa having only a fraction (313K of 10 million) of its collections on GBIF. I searched on GBIF for some herps that I know are endemic to DR Congo and found zero locality records, underscoring this point. My colleague Werner Conradie has done a lot of recent herp work in the Okavango Delta, one of the areas flagged as poorly explored, but I do not see the Port Elizabeth Museum (South Africa) collections listed on GBIF.
Line 360: check grammar “vastness of Russia too seems undersampled”
Lines 378-379: check for consistent use of straight vs curly quotation marks
Line 388: the comments about lack of infrastructure and permits is valid, but because this is the first time it seems to be mentioned, perhaps it belongs in the Intro or Discussion. The authors might also want to refer to this paper, which brings up similar issues:
Britz, Ralf, Anna Hundsdörfer, and Uwe Fritz. 2020. “Funding, Training, Permits—The Three Big Challenges of Taxonomy.” Megataxa 001 (1): 049–052.
Line 393: just a suggestion, but the authors might want to check out this book, which talks about marginalized indigenous knowledge- my take is that the book doesn’t really talk about science (and nothing at all about collections), but it still has useful information that might be relevant to this paper:
Hernandez, Jessica. 2022. Fresh Banana Leaves: Healing Indigenous Landscapes Through Indigenous Science. Berkeley: North Atlantic Books.
I completely agree that parachute science should be avoided and indigenous knowledge should be incorporated into future efforts, as long as the misguided idea of “compassionate collection” is rejected (see Nachman et al. 2023). But again, without a more expansive discussion of how “unexplored” has been used by others, I can’t say whether I agree that its use is harmful, or if “biodiversity blindspot” is a less harmful term. Regardless of what the authors decide to do here, however, I encourage them to emphasize that all collections should be digitized and merged into GBIF, so that the G (global) in the title truly becomes a reality. Because once this is done, and the global scientific community can access ALL data, the “harmful” connotations of unexplored will be a moot point. I would argue that unexplored or poorly explored regions could then be identified and mapped with even more precision, which would prioritize future collection efforts and fill in these gaps. Food for thought.
Line 419: delete period after West, and if the point about this sentence is about the wealthy West, why are examples from the developing world?
Line 429: I got lost about the comment regarding perpetuating colonialism. How is this related to digitization of collections? Instead of focusing on political correctness here, why not call for more funding to help collections in developing countries with no resources to digitize their collections to GBIF?
I do not wish to remain anonymous: Eli Greenbaum

Experimental design

See section 1.

Validity of the findings

See section 1.

Additional comments

Feel free to ignore this, but a few additional thoughts:

I confess I have used the terms “poorly explored” and “poorly known” to describe knowledge of terrestrial biodiversity in Central Africa, especially for herps (Emerald Labyrinth book). In the same book, I also note that biodiversity surveys from the lowlands of DR Congo are more difficult because of a lack of infrastructure (poor roads, collapsed bridges, etc.) and many deadly tropical diseases. In the case of reptiles, the authors in the following paper agree that Central Africa is a huge blind spot:
Tolley et al. 2016. Conservation status and threats for African reptiles. http://dx.doi.org/10.1016/j.biocon.2016.04.006

Reviewer 3 ·

Basic reporting

Yes. Few grammatical errors (see below in Additional comments).

Experimental design

Yes.

Validity of the findings

Not quite, the authors present the results as if they are true biodiversity under sampled areas, yet this is largely ignoring any natural history collections who currently do not have their data digitized and online for the public. See Additional comments below.

Additional comments

Review of What “unexplored” means: Mapping undersampled regions in natural history collections.

This manuscript examines the lack of biodiversity data in natural history collections (via GBIF) from around the world, for fungi, vascular plants (terrestrial and freshwater), and vertebrates to highlight areas lacking in biodiversity data. The authors discuss what “unexplored” might actually mean and recommend retiring term for more nuanced phrasing to mitigate future misunderstandings of natural history science.

While this is an interesting and important topic for biodiversity science, conservation biology, climate change observations, as well as many other fields, their results may better reflect the number of “digitized” museum collections around the world (a phenomena they are aware of and discuss to limited extent), biodiversity hotspots, and regions lacking diversity, rather than what they anticipated, which areas are truly unexplored (not represented by scientific collections). I think it could be an important contribution to the field of natural sciences, but it needs some rewording and refocusing to better represent the results.

For instance, I would recommend changing “undersampled” to “under represented” in the title (line 2), because they acknowledge that many institutions do not yet have their data digitized (and therefore, not on GBIF), meaning areas covered by those institutions may have been “sampled”, they just have not been represented on public databases (such as GBIF).

Secondly, I don’t see a need to eliminate the term “unexplored,” as long as users of that term define what they mean when they are using it. This is suggestive of a “woke” community trying to “correct” previous faux pas in western science. Instead, the authors recommend the term “blind spot” for areas that have not been digitized and aggregated into publicly accessible databases (lines 50–51; 394–396). To me, the term “blind spot” means something overlooked, or not seen because of obstacles, whereas the lack of digitization and inclusion in aggregate databases can be the result of several other things, such as lack of funding (which they acknowledge), lack of staff (related to funding). However, some institutions may choose not to make their data publicly available for various reasons, and therefore the data are not in aggregate databases. Thus, museum staff may be aware of this issue, they just do not have the staff, funding, or desire to prioritize this paucity. I would not consider these reasons “blind spots” in terms of biodiversity being surveyed or not. “Blind spots” to the quick data-grabbing GBIF users perhaps (I say this as a user of GBIF).

If the authors see problems with the use of the term “unexplored,” perhaps they should provide some examples where it was used in these mis-leading ways, and the consequences that have come of it. They provide no actual examples; therefore, this could be considered a “straw man argument.” Again, they recommend usage of the phrase “biodiversity blindspots” (lines 433–436) for places “poorly-collected” and “undigitized”. Why not have separate terms for these very different phenomena? Maybe “blindspots” for undigitized, but I don’t see the need for, nor a better term for places that remain unexplored (not visited/collected by scientists).

I also see some problems in the wording of their conclusions. The term “exploited” (lines 380, and 399) has very negative connotations and perpetuates this idea for science and natural history collections. This invokes a connection between un-explored and un-exploited. Yes, in earlier, colonial times, many countries were exploited, as described by the following concept of “parachute science,” but more recent efforts have been made to rectify this behavior, many institutions are returning sensitive collections to countries that want them back and can care for them. And many scientists today try to include local researchers when available, either for permitting purposes or legitimate intentions for capacity building, or both. And with particular regard to the genetic resources and the Nagoya Protocol, any signing country can create their own rules and regulations with regard to foreigners collecting in their countries. I think the use of exploitation in this regard perpetuates a negative connotation for scientists and natural history collections.

The best results of their data are the lack of digitized collections. To me, their maps represent global biodiversity as represented by digitized collections. If they were truly mapping areas where scientists have not made natural history collections, they should also include data from institutions that are not yet digitized, for whatever reasons. They mention a few institutions that likely have large collections from areas showing paucities, but if they truly which to map “unexplored” areas, they should conduct a little more research into these institutions and their holdings and how that would change their results. Additionally, they should conduct more thorough searches of natural history museums with large holdings that are not yet digitized.

Their choice of taxa seems a bit odd and could use more explanation (vertebrates and plants, when invertebrates make up more of the earth’s biodiversity). Is this an issue of these collections being more likely to be digitized? Or these are the taxonomic group with which the authors are familiar? I think a little explanation is warranted.

There is also the issue of areas being natural more diverse, and other areas less. They mention this, lines 45–47. However, no measures were made to correct for this phenomenon in the data, as far as I can tell. For instance, northern Canada has way fewer species in that region than elsewhere in NA. Do they expect that scientists would collect more individuals of fewer species in areas of low diversity? Probably not, but this is undoubtedly a difficult issue to deal with. How many species occur in each region, and we expect X number of specimens to be collected. A ration could be used to “standardize” their results (e.g., the number of expected species per polygon, and a number of specimens of each species). At least some mention of this issue should be addressed in more detail.

I appreciate them defining digitization, because I think a lot of institutions have come to use this term for making digital images of their specimens available, rather than transcribed metadata.

Other minor issues, comments, suggestions,

Line 121, v.x.x ? is there version that should be included, or is it supposed to be “x.x.”?

Line 129 (and line 230), do they mean Gulf of California? And why not “islands”? I was not aware of many “landmasses” in the Gulf of Mexico housing significant biodiversity. Are these rocky reefs inhabiting shore birds?

Line 148 should use n-dash for 1–5.

Lines 164–67, they say that some museum might list “Brazil” as a locality for older specimens, and the problem of the centroid being in the middle of the country. Then, they state they “avoid descriptions of such artifacts here.” Does that mean they included them, but don’t discuss, or they removed them from their analyses?

Line 193, “Basin” should be capitalized (Orinoco Basin).

Line 214 Remove “that we examined” unless, there exists a larger group of vertebrates that they didn’t examine.

Line 216, “Basin” should be capitalized (Amazon River Basin).

Lines 257 and 259, “Bird” and “Fungi” should not be capitalized (respectively).

Lines 266–273. I found this part very confusing, perhaps it could be rewritten to be more clear. First, lines 266–271 contain a large, run-on sentence. This should be broken up. Is the MBG on Madagascar proper? (…the “large island”…). And it is unclear on which island local knowledge would be hindered.

Lines 275–76. I don’t see herps being different here, say from any other vertebrate group. In fact, herps and mammals seem most similar in this region (“diagonal band tropical southern equatorial regions”).

Line 419, remove period after “West.” (….).”

---

## Round 0.2 · Minor Revisions

As you can see, the reviewers have only minor suggestions. Please fix them and resubmit. I expect to accept the paper as soon as you do.

Reviewer 1 ·

Basic reporting

I really appreciate authors attended the recommendations and expanded their bibliography to include relevant information.

Experimental design

I think authors improved very much their methods by carefully curating GBIF data. I recognize their effort in providing more statistical robustness by using the Gi stats. I just wonder if this statistics has been used on biodiversity studies in the past.

Validity of the findings

The relevance of the study is evident. Authors provide, with detail, areas that require increased research and collection efforts due to lack of specimens with vouchers and call for a careful definition of unexplored areas in terrestrial habitats.

Additional comments

Authors have greatly improved the manuscript and I truly appreciate their willingness to follow reviewers suggestions.
Just one minor suggestion and authors are free to take it or not:

I am not fully convinced on how the authors want people stop using the word unexplored. I understand their means, but I think a better way to do this is by encouraging a better use of the word unexplored. For example, in microbiology there is still a lot to explore and discover (see https://www.nature.com/articles/s41564-024-01686-x); so, in this context, unexplored seems a valid concept.

When the authors say

´we call for the disuse of the term "unexplored" which is frequently used both by academic and non-academic sources to mean an area previously uninvestigated or poorly known to Western Science,´

they reference the use of unexplored mostly in a context of tourism, with just one old reference regarding to natural history. I don't clearly see how the authors show the use of "unexplored" in modern times to refer to areas lacking collections with vouchers. That is why I suggest authors to simply advocate for the correct use of the word unexplored.
Authors may find also support on the following paper: https://www.pnas.org/doi/10.1073/pnas.2022218118, which clear things out on the importance of local indigenous knowledge on contributing biodiversity knowledge and how most places on land have been already explored but lack of infrastructure and resources, particularly in global south, make them appear as poorly known.

I love the idea of blindspots as those areas where we have not --or are not sure we have-- enough information of specimen collections. See for example this recent article about dark spots: https://www.theguardian.com/environment/2024/oct/01/kew-botanic-gardens-study-33-dark-spots-plant-species-identification-unknown-biodiversity-

Congrats for the efforts.

Reviewer 3 ·

Basic reporting

Review of: What ‘unexplored’ means: Mapping regions with digitized natural history records to look for ‘biodiversity blindspots’ for the journal PeerJ.

This is my second review of this manuscript. The authors have improved the manuscript based on recommendations of reviewers. The data are somewhat “cleaner” after their filtering and they have tidied up some of the language. However, the authors still seem to “double down” on this replacing the term “unexplored” with “blindspots” for all cases of places on earth not represented by digitized data occurrences of voucher specimens of their selected taxa (terrestrial vertebrates, freshwater fishes, plants and fungi) despite the fact that myself and Rev. #2 raised this issue of two types of situations. This seems to be the driving force of the manuscript; I don’t agree with this suggested replacement of terms as it is presented (one catchall for another, see below). However, I do believe the authors raise awareness to several issues that are worth sharing with the community (though, some of it is already known).

The authors use publicly available, online museum data (of their selected taxa) from the data aggregator GBIF to assess what areas of the world are “unexplored,” and what it means to be “unexplored.” They acknowledge that areas lacking online voucher specimens might be because museums might have data from those regions but they are not yet digitized and refer to these as “biodiversity blindspots” and call for museums to digitize their data. Two issues, unexplored=blindspots and museums should digitize their data; addressed separately below.

Blindspots. Lines 37–42, 93–98, and 373–392 (of the pdf, throughout my review)

A point raised by myself and Rev. #2 in the previous version, that the authors recommend replacing the term “un-explored” with “blindspots.” In the previous version, two reviews pointed out there were no reference to people using the term unexplored. They have provided several references; however, they don’t explain how the use of that term is harmful, save for the Conclusions section (Lines 386–392, addressed further below). They point out that the causes maybe be not studied by Westerns, un-inhabited, remoteness, lack of diversity, few organisms (e.g., Sahara), and then political issues (unrest), lack of infrastructure, for permits, and other areas just plain overlooked (Lines 373–386).

These reasons are very different: being unexplored, or explored, but data not publicly available. Myself and Rev #2 found this dichotomy apparent, acknowledge it, and at least myself recommend keeping “unexplored” (for the areas not visited/collected by Westerners). However, the authors, acknowledge this dichotomy (no data vs. data, but not publicly available), but seem to insist on using the term “blindspots” to cover all of these differences. I previously suggested, and continue to recommend the use of “blindspots” for “data, but not publicly available.” I find this reminiscent of the “known” and “unknown” terms of dark taxa discussion in the literature (see Collins & Cruickshank, 2014 Syst Biol 2014 Nov;63(6):1005-9. doi: 10.1093/sysbio/syu060). I think it would be useful to have a term to describe the latter, “the known, unknowns” if you will (e.g., blindspots). However, I don’t find it useful to replace one catchall term with another. I would argue the areas “just plain overlooked” are truly “unexplored” and the use of that term appropriate. It would even better to have other terms for different phenomena, such as “un-accessible,” etc.

I don’t see how the term “unexplored” is harmful, as described by the authors (Lines 386–392). They state it:
“…may inadvertently perpetuate misconceptions, or undermine the value of existing research and the efforts of local scientists and indigenous people or disregard the underlying resource-imbalances
impeding natural history research, collection, and digitization in many regions.” (Lines 386–392)

Yet, they don’t provide examples nor references to these points. Further, I don’t see how it referring to areas of local scientists not yet unified with the greater scientific community as “blindspots” makes it any better (what would the blind think of this?), particularly if it pools them with other categories with a lack of information in the region for different reasons. These are “unknown, unknowns” in that regard. The value is identifying local scientific collections not yet unified with the greater scientific community (through paucities in data, but what about those in areas seemingly “covered”, they likely also contain unknown unknowns) and willingly bring them into the greater scientific community (or make their existence known). And again, it may identify areas limited by resources, and that is yet another separate issue discussed further below.

The point is, there are many factors that may contribute to unexplored regions of the world, yet replacing one catchall term with another is not particularly useful. What is useful is identifying the different causes, for lack of data: Known and has collections not yet digitized (see below), Unknown, collections exist but remain unknown to the greater scientific community, Unknown, not visited by natural history collectors.

The strongest point of the paper in my opinion may be the new Table 1, showing the skew of where collections are housed and where they are from. What would be even better to see would be this compared with an estimate of biodiversity.

Museums should digitize their data. Lines 40–44, 413–425.

I think most museum staff are aware of this issue. Large initiatives have been made around the world for digitization of museum data, particularly for the groups of interest to the authors, and most major collections have been digitized and made publicly available (e.g., the various contributors to VertNet). There may have been some institutions that for various reasons chose not to make their data publicly available. Further, there may be smaller regional collections that were not included in the initial, broad attempts to digitize natural history data and that lack funding and staff. The authors state that some “West” institutions with relatively few (<15%) of their collections in GBIF” citing Smithsonian NMNH (lines 420–423). I assume the authors are basing this off of the number of objects from the institution’s collections website; 148 million versus the number in GBIF (~9 million). However, if one looks more closely, NMNH has the vast majority of terrestrial vertebrates digitized (over 414,000 amphibians, 179,000 squamates, 627,000 birds, 681,000 mammals), closer to 95% of their holdings listed on their websites. The fishes and other groups (e.g. tadpoles) may be classified as a single “voucher” when in fact they may be composed of “lots”, series of contemporaneously collected individuals (“objects”) presumed to be the same species. Furthermore, maybe they are not aware of the 1,000s of large specimen cabinets, each full of trays of pinned insects with tiny hand-written tags containing the collection data, that go back decades, or over a century in time (33 million specimens, only 722,000 digitized records; https://collections.nmnh.si.edu/search/ ).
This and other objects (minerals, fossil specimens, etc.) likely make up the huge discrepancy between the number of objects boasted and the number of vouchers in GBIF. Of course, the entomology staff would be grateful if all of their specimens were digitized, and some museums are working with conveyor belt scanners and optical character recognition software to digitize these data, we are still a way from having that technology in practical use. Until then, staff are just trying to keep up with the day-to-day loan requests, new acquisitions and collection maintenance. So, a “call for museums” to digitize their data may fall flat on many institution staff. Unless you are proposing a new project to obtain a funding source, this has already been heard by many. Have any of the authors been to TDWG or SPNHC meetings? I think what would be better would be to point out these short-comings and potentially offer new avenues for overcoming them. Perhaps a website or database that identifies the institutions holding specimens but lacking infrastructure to digitize.

What gives me more concern are the details in these cases and making broad generalizations with GBIF data. Reviewer #1 pointed out some issues with their searchers and they seemed to have made corrections to mitigate biases. Reviewer 2 pointed out an additional museum with collections in Africa, and I’ve mentioned the discrepancies in their expected results from NMNH, but how many other museums are out there with data in “blindspots” or other issues? Again, I think it may be premature to obtain accurate results for these broad-level questions. Given that they have cited others doing similar things (using the term “unexplored”) and other broad generalizations, perhaps a better contribution would be to alert users to be more cautious in interpreting their results from broad sweeping studies (of higher taxonomic groups) from GBIF records alone and the need to carefully comb through the data to make certain it represents what they expect (see lines 60–62).

Other Minor issues.

Previously, I raised the concern that their study does not take into account species diversity. For instance, in both versions, they make comments about Canada and northern central USA lacking records for amphibians and reptiles (Lines 230–233). Yet this area generally lacks species diversity for herps. Again, do they expect that researchers would collect more of the same species to compensate for the lack of collections? I’m not sure how log-transforming the data accounts for this?

Nicaragua should probably be put in the category with Cuba and Venezuela (political issues), as it has been historically difficult to work in as well. Lines 218–221.

Lines 60–62. GBIF should probably be spelled out here (first use)? Or, just use “aggregate databases” here and below in the Mat-Methods (lines 101–102) say “…from the *aggregate database* GBIF…”

Experimental design

See above

Validity of the findings

see above

Additional comments

none.

---

## Round 0.3 · accepted · Accept

Thank you for your prompt revision — and for submitting this to PeerJ!